# Radiomics for the Prediction of Overall Survival in Patients with Bladder Cancer Prior to Radical Cystectomy

**DOI:** 10.3390/cancers14184449

**Published:** 2022-09-13

**Authors:** Piotr Woźnicki, Fabian Christopher Laqua, Katharina Messmer, Wolfgang Gerhard Kunz, Christian Stief, Dominik Nörenberg, Andrea Schreier, Jan Wójcik, Johannes Ruebenthaler, Michael Ingrisch, Jens Ricke, Alexander Buchner, Gerald Bastian Schulz, Eva Gresser

**Affiliations:** 1Department of Diagnostic and Interventional Radiology, University Hospital Würzburg, Würzburg-Oberdürrbacher Str. 6, 97080 Würzburg, Germany; 2Department of Urology, University Hospital, LMU Munich, Munich-Marchioninistr. 15, 81377 Munich, Germany; 3Department of Radiology, University Hospital, LMU Munich, Munich-Marchioninistr. 15, 81377 Munich, Germany; 4Department of Radiology and Nuclear Medicine, University Medical Center Mannheim, Mannheim-Theodor-Kutzer-Ufer 1–3, 68167 Mannheim, Germany; 5Department of Otolaryngology, University Hospital, LMU Munich, Munich-Marchioninistr. 15, 81377 Munich, Germany; 6Faculty of Medicine, Medical University of Warsaw, Żwirki i Wigury 61, 02091 Warsaw, Poland

**Keywords:** bladder cancer, radical cystectomy, radiomics, outcome prediction

## Abstract

**Simple Summary:**

Accurate prognostic assessment of bladder cancer patients is essential for risk stratification, individualized therapeutic decision-making and follow-up management. In this study, the potential of quantitative features from preoperative CT images (radiomics features) to predict overall survival in patients treated with radical cystectomy was investigated. Both bladder tumors and pelvic lymph nodes as well as their immediate surroundings were segmented and analyzed. Regression models based on radiomics and clinical parameters were developed and compared. The combination of radiomics features from all regions with clinical parameters achieved the best results with a mean area under the ROC curve of 0.785 integrated over 1 to 7 years after radical cystectomy. Furthermore, the combined model stratified patients into high- and low-risk groups with significantly different outcomes. Therefore, the prognostic information from preoperative CT images could aid in the early stratification of patients with bladder cancer even before RC is conducted and could complement the well-established clinical factors.

**Abstract:**

(1) Background: To evaluate radiomics features as well as a combined model with clinical parameters for predicting overall survival in patients with bladder cancer (BCa). (2) Methods: This retrospective study included 301 BCa patients who received radical cystectomy (RC) and pelvic lymphadenectomy. Radiomics features were extracted from the regions of the primary tumor and pelvic lymph nodes as well as the peritumoral regions in preoperative CT scans. Cross-validation was performed in the training cohort, and a Cox regression model with an elastic net penalty was trained using radiomics features and clinical parameters. The models were evaluated with the time-dependent area under the ROC curve (AUC), Brier score and calibration curves. (3) Results: The median follow-up time was 56 months (95% CI: 48–74 months). In the follow-up period from 1 to 7 years after RC, radiomics models achieved comparable predictive performance to validated clinical parameters with an integrated AUC of 0.771 (95% CI: 0.657–0.869) compared to an integrated AUC of 0.761 (95% CI: 0.617–0.874) for the prediction of overall survival (*p* = 0.98). A combined clinical and radiomics model stratified patients into high-risk and low-risk groups with significantly different overall survival (*p* < 0.001). (4) Conclusions: Radiomics features based on preoperative CT scans have prognostic value in predicting overall survival before RC. Therefore, radiomics may guide early clinical decision-making.

## 1. Introduction

Bladder cancer (BCa) is the most common malignancy of the urinary tract. About one out of 100 men and one out of 400 women will be diagnosed with BCa in the course of their lives [1,2]. Radical cystectomy (RC) with bilateral pelvic lymph node dissection is the reference standard for patients with localized muscle-invasive tumors in curative intent. BCa is a heterogeneous disease with the majority of patients presenting with superficial non-muscle invasive tumors with a 5-year survival rate of over 90%, although recurrence rates are high. Survival rates drop drastically as the tumor invades different layers of the bladder to less than 10% for distant disease spread [3,4,5,6,7]. However, patients within the same tumor stages show substantial individual variability in survival outcomes. Risk factors such as age at RC, pT- and pN-status as well as lymphovascular invasion and positive surgical margins are associated with a bad outcome prognosis [7,8,9,10,11]. Despite the increasing understanding of the underlying pathophysiology of BCa and the use of (neo)adjuvant treatment strategies, mortality rates remain high and the oncological outcomes following RC have not significantly improved over the last few decades [10]. It has also been shown that the disease-specific mortality is highest during the initial years of follow-up after RC and decreases for survivors over time. Better estimates of survival probabilities early after RC or even in advance of treatments may lead to optimized patient monitoring [12]. The development of models that can accurately determine overall survival—especially within the first years of follow-up after RC—might support oncologists in devising proper treatment and follow-up strategies for the individual patient [13,14]. Radiomics, which uses a large number of quantitative parameters extracted from imaging data to predict clinically significant diagnostic or prognostic variables, might be a promising field in this context [15]. This study evaluates radiomics-based models to predict overall survival in patients with BCa after RC based on preoperative CT imaging. It also evaluates an approach combining these imaging features with well-established clinical risk factors.

## 2. Materials and Methods

### 2.1. Study Design and Cohort

From 1345 consecutive patients screened in our database, a total of 301 patients with urothelial or squamous BC who received RC in our hospital with pelvic lymphadenectomy between February 2004 and March 2021 were included in the study. Patients underwent complete preoperative CT of the pelvis in a venous contrast phase within 3 months before the operation. Additionally, complete postoperative histopathological reports for the primary tumor and lymph nodes were extracted from our database. The exclusion criteria were (I) primary tumor other than urothelial or squamous cell BCa, (II) previous systemic treatment for BCa (neoadjuvant chemo- or radiotherapy), (III) secondary malignancies, (IV) incomplete CT imaging within 3 months of RC or incomplete pathological reports, (V) impossible or insufficient primary tumor segmentation or (VI) loss to follow-up (Figure 1). All patients were treated with curative intent. Clinical and histopathological parameters including age, sex, pT- and pN-stages, lymphovascular invasion of the primary tumor, and positive surgical margin at RC were extracted from medical records. All patients in the study cohort were followed up after three months, twelve months and then annually by postal surveys, as follow-up examinations were partly performed in our institution but also in urological outpatient clinics. This retrospective single-center study was approved by the local ethics committee (Ethikkommission bei der LMU München, protocol code 18-459, 2 August 2018).

### 2.2. Segmentation and Radiomics Feature Extraction

CT scans were acquired from various scanners using differing protocols, at ours or external institutions (Appendix A). Segmentations of the primary bladder tumors as well as all peri-iliac, obturator, and perivesical LNs on both sides were performed manually on axial CT images in the portal venous phase using the open-source Medical Imaging Interaction Toolkit (MITK, version 2018.04.2, DKFZ, Heidelberg, Germany). The segmentations were subsequently reviewed and corrected, if necessary, by a second reader with extensive experience in urogenital imaging. Both readers were blinded to the clinicopathological information. In the following, the segmentations including the area of the region of interest (bladder tumor or lymph node) are referred to as intratumoral masks. Peritumoral masks, comprising the area within a 6 mm margin around the intratumoral masks, were automatically created from manual segmentation using morphological operations from the SimpleITK library. First-order and shape features (*n* = 33) were extracted from all segmentations of the primary bladder tumor as well as from the largest segmented lymph node from the intra and peritumoral masks (132 features in total). For feature extraction, our in-house framework, so-called AutoRadiomics, was used [16], (Appendix A).

### 2.3. Model Development

The patient cohort was split into a training (80%) and a test (20%) cohort. A Cox proportional hazard model with an elastic net penalty was used, as implemented in the scikit-survival library [17]. Two parameters, L1 ratio and alpha, were optimized using grid search in the training cohort in the setting of a 5-fold cross-validation (Appendix A). The selected features in the final models can be found in Appendix A. The Brier score, integrated over 1 to 7 years after surgery, was used as a metric for parameter optimization. The Brier score, which is equivalent to the mean squared error applied to probabilities, was selected to guarantee good calibration of the model’s predictions. The study design is demonstrated in Figure 2.

### 2.4. Combining Radiomics and Clinical Parameters

As a reference, a model using only the clinical parameters, including age, sex, pT- and pN-status, the presence of residual tumor in the surgical margins, and the presence of lymphovascular invasion by the primary tumor, was trained. Moreover, several radiomics-based models and a combined model of radiomics features and the named clinical parameters were evaluated.

### 2.5. Evaluation Metrics

The area under the receiver operating curve (AUC) and the Brier score, both integrated over the time period from 1 to 7 years after surgery, were used as the primary scoring metrics. The AUC for survival data quantifies how well a model can distinguish subjects who experience an event before a given time point from those who do not. The Brier score evaluates the accuracy of probabilistic predictions. The lower the Brier score, the better the predictions are calibrated. An integrated metric (AUC and Brier score) refers to the mean metric value over a time range. Additionally, Harrell’s concordance index (C-index) was reported. The C-index corresponds to the proportion of all comparable pairs in which the predictions and outcomes are concordant.

The median risk score in the training cohort was used as the threshold to dichotomize patients into low-risk and high-risk groups, which were compared using Kaplan-Meier analysis. The calibration of the combined model was assessed by plotting the predicted survival probability against the observed survival rates at 1-year, 3-year and 5-year time points. The period from 1 to 7 years after surgery was defined a priori based on the median follow-up time and clinical experience to capture the variability in outcomes.

### 2.6. Statistical Analysis

Differences in the clinicopathological variables between training and test cohorts were compared using Welch’s t-test, Chi-square test for independence, or Fisher test, wherever appropriate. Confidence intervals of 95% (95% CI) were calculated with the bootstrap technique, using 1000 resamples (with replacement) of predicted probabilities to determine the CI. A log-rank test was used to compare high-risk and low-risk groups. P-values for integrated AUC and Brier scores were calculated using the distribution of differences from bootstrapping with 1000 resamples. A *p*-value < 0.05 was considered the threshold for statistical significance. All statistical analyses were implemented in the programming language Python (version 3.10, Python Software Foundation, Wilmington, DE, US). The code used for our analysis will be shared after publication.

## 3. Results

### 3.1. Patient Characteristics

Of the 1354 patients screened, 301 were included in the final study cohort. A total of 240 patients were randomized into the training cohort and 61 into the test cohort. In total, 1178 lymph nodes were segmented in the training and 296 in the test cohort. The detailed clinical patient characteristics of the training and test cohorts are presented in Table 1. No statistically significant differences were found between the training and test cohorts regarding age, sex, pT- and pN-stages, tumor volume, surgical margins, lymphovascular invasion, the time between preoperative CT and RC, as well as the time to recurrence. In the analyzed cohort, the median follow-up time was 73 months (95% CI: 56–85 months) and the median survival of the overall cohort was 56 months (95% CI: 48–74 months).

### 3.2. Survival Prediction

The primary results, including integrated AUC and Brier, as well as C-index, are presented in Table 2. The combined model of clinical characteristics and radiomic features from intratumoral and peritumoral regions yielded the best performance in terms of an integrated AUC of 0.785 (95% CI: 0.648–0.891) with an integrated Brier score of 0.175 (0.129–0.224) and a C-index of 0.740 (0.609–0.81). The clinical reference model achieved an integrated AUC of 0.761 (95% CI: 0.617–0.874) in the test cohort with an integrated Brier score of 0.185 (0.144–0.232) and a C-index of 0.722 (0.609–0.823). When the radiomic features from peritumoral regions segmentations of the primary tumor and LN segmentations were additionally included, the model achieved an integrated AUC of 0.771 (0.657–0.869) with an integrated Brier score of 0.202 (0.163–0.243) and a C-index of 0.737 (0.644–0.836). The time-dependent AUC over the time range from 1 to 7 years after RC is presented in Figure 3 for the clinical and radiomics models. Table 3 further distills the accuracy of survival predictions at discrete time points of 1, 2, 3, 5, and 7 years after RC. The Kaplan–Meier analysis of the test cohort, stratified into high- and low-risk groups by the combined clinical and radiomics model, is presented in Figure 4. The combined model was able to dichotomize the cohort according to the risk score with significantly different overall survival (*p* < 0.001). The comparative Kaplan–Meier analysis for the radiomics-based model and the clinical model, as well as a model based on the TNM classifications, is shown in Appendix A.

### 3.3. Prediction Calibration and Interpretability

Figure 5 presents the calibration curve for the combined model at 1-, 3- and 5-year time points (A), as well as the most important features (B). It can be seen that the predicted probabilities closely correspond to the true survival proportions. Already verified clinical risk parameters such as high pT status ≥3, pN status, and positive surgical margins (R1 status) as well as age were of high relevance. Moreover, several radiomic first-order and shape features, including the median voxel intensity within the primary tumor (Median) and the volume of the primary tumor (VoxelVolume), as well as intra and peritumoral features from the lymph node masks (Energy and Variance), were among the most predictive features.

## 4. Discussion

This study evaluates clinical and radiomic features from preoperative CT scans for the prediction of the overall survival of patients with bladder cancer following radical cystectomy with curative intent. A radiomics model based on intra- and peritumoral segmentations of the primary tumor and lymph nodes reached a similar prediction performance with an AUC of 0.771 (0.657–0.869) to the validated clinical parameters with an AUC of 0.761 (0.617–0.874), integrated over the time range of 1–7 years of follow-up after RC. Combining clinical parameters and radiomic features yielded the best performance in terms of an integrated AUC of 0.785 (0.648–0.891). Patients could be stratified into high-risk and low-risk groups with significantly different outcomes. Additionally, the overall survival of patients for the annual follow-ups in the first to seventh years after RC was evaluated, with an AUC ranging from 0.722 to 0.825 for the radiomics model versus an AUC ranging from 0.692 to 0.819 for the clinical model. In this study, the use of radiomics parameters for the overall survival prognosis in patients with BCa after RC achieved substantial performance, comparable to models based on validated clinical risk factors. In contrast to parameters of the clinical model, which are mostly accessible after RC, imaging features for radiomics survival prognosis can be collected non-invasively before operative treatment has been conducted. Our results suggest that radiomics might have a relevant additive value in the outcome prediction of BCa patients.

Despite modern advancements in surgical techniques and post-operative treatment options, outcomes of BCa patients after RC remain poor, with 5-year overall survival rates of around 60% [11,18]. This is reflected in our study, with a median survival of 56 months within the study cohort. Clinical decision-making regarding the follow-up scheme and post-operative adjuvant treatment regimen markedly depends on the TNM staging, not taking into account other risk factors that might play an important role in outcome prognosis after RC. Therefore, further models facilitating outcome prediction are of high importance. Several clinical parameters have been identified as important prognostic factors, including age, pN-status, pT-status, tumor grade, lymphovascular invasion and positive surgical margins [7,19]. In recent years, an increasing number of machine learning models and clinical nomograms have been evaluated for the prediction of disease-specific and overall survival in BCa patients, mostly based on clinical and epidemiologic data from medical reports. Reported AUCs for prediction of 5-year survival ranged up to 0.81 [20,21,22]. One study used information from medical records combined with features extracted from whole-slide immunofluorescence images to identify high-risk and low-risk groups in terms of 5-year prognosis with an AUC of 0.89 from combined models [23].

CT imaging has a fundamental role in the diagnosis, staging, treatment guidance, and response monitoring in BCa [3]. As imaging-based features can be easily obtained non-invasively at low costs, radiomics has been increasingly evaluated for initial detection, grading, and local as well as nodal staging of BCa patients [24,25,26,27,28,29,30], for recurrence assessment [31,32,33] and for the response evaluation to neoadjuvant preoperative as well as adjuvant treatments in recent years [34,35,36,37,38,39,40]. However, to our knowledge, no imaging-based machine-learning models have been investigated for outcome prediction of BCa patients before RC. Our study evaluates a radiomics approach to assess overall survival within the first years of follow-up after RC. Moreover, an automated peritumoral segmentation to include the information from the tumor margin and the surrounding soft tissues was applied. The inclusion of the peritumoral segmentations had a positive predictive impact on the models’ performances. The interest in the role of the tumor microenvironment in the pathogenesis of BCa is rising [41]. Peritumoral segmentations might therefore capture important additional information from the surrounding stroma and were among the most relevant features chosen for the model’s calibrations. The approach of peritumoral radiomics feature extraction has already been positively evaluated for outcome assessment in other tumor entities such as glioblastoma [42], hepatocellular carcinoma [43,44] and lung cancer [45].

Currently, adjuvant therapy schemes are substantially changing as the field of immunotherapy is emerging for BCa patients after RC. Using immunotherapy for adjuvant therapy in muscle-invasive BCa has demonstrated promising results and has recently reached approval, thereby also addressing poor responders to conventional chemotherapy as well as patients that could not be considered for conventional chemotherapy [46]. Therefore, a better understanding of outcomes for individual patients is of increasing importance to propose a risk-based strategy in surveillance and postoperative treatment options. In this context, imaging-based radiomics approaches might help guide optimal treatment options for patients after RC.

### Limitations

The retrospective design of the study, the limited sample size and the use of multiple scanners for CT scan acquisition are the main limitations of this study. In addition, the algorithms developed in the present study have not been validated externally. In order to successfully apply these models in the clinical setting, future studies need to verify the generalizability of the imaging features and standardized parameters. Moreover, the effect of neoadjuvant treatment on overall survival cannot be assessed in this study due to the exclusion of this patient subgroup. Additionally, the effect of adjuvant chemotherapy after RC was not evaluated in this study. Results of a meta-analysis testing for the prognostic value of adjuvant chemotherapy indicated no effect on survival prognosis and no level-1 evidence has yet demonstrated a significant survival benefit to BCa patients after RC [7,47].

## 5. Conclusions

Radiomics may support the survival stratification of BCa patients non-invasively before RC and could be assessed to guide therapeutic decision-making and follow-up management.

## Figures and Tables

**Figure 1 cancers-14-04449-f001:**
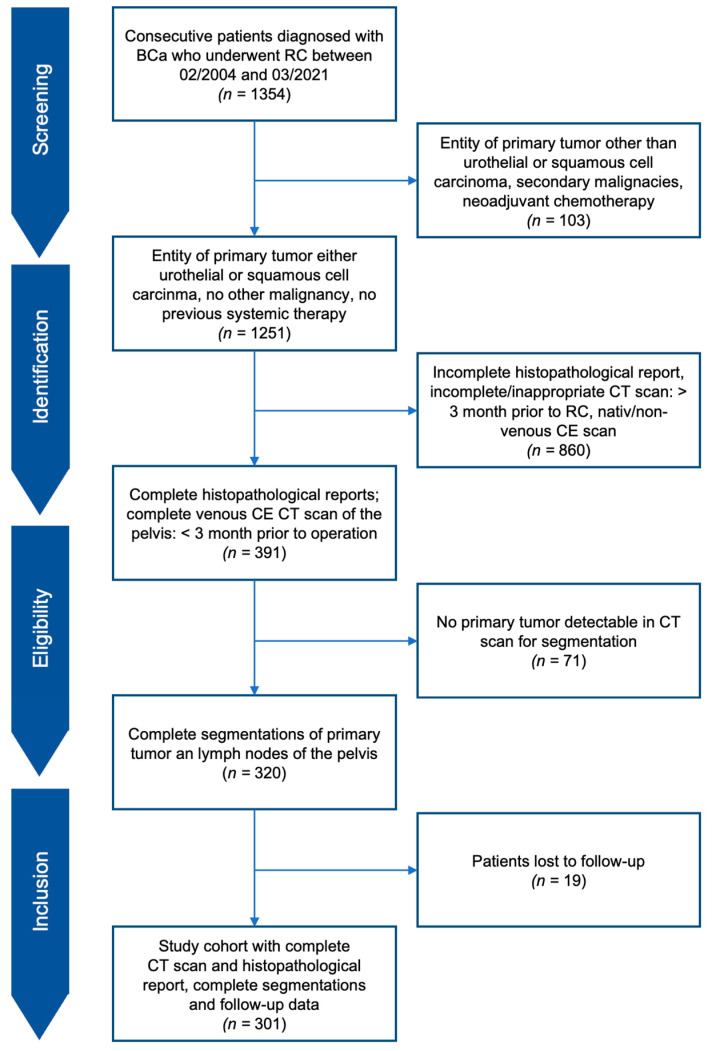
Study flow chart.

**Figure 2 cancers-14-04449-f002:**
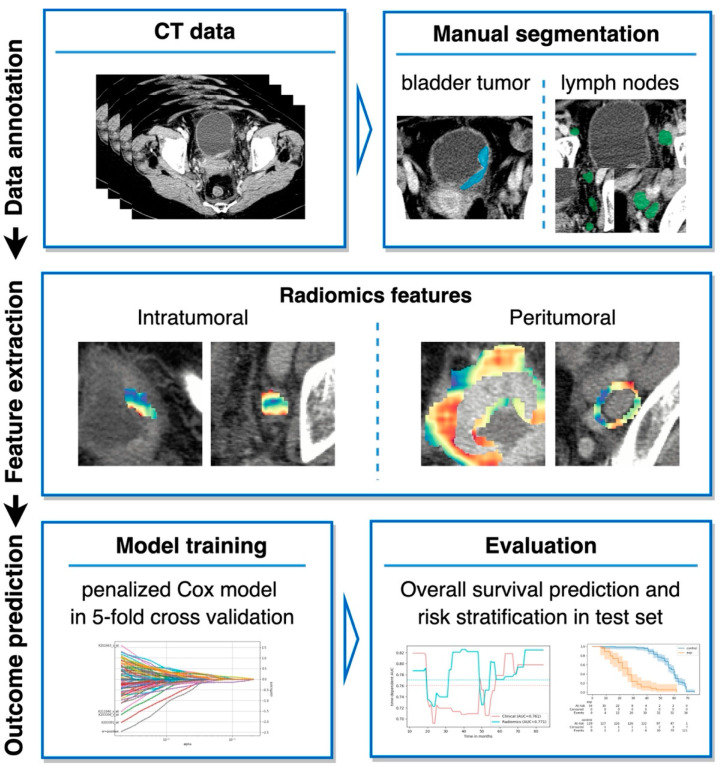
Study design.

**Figure 3 cancers-14-04449-f003:**
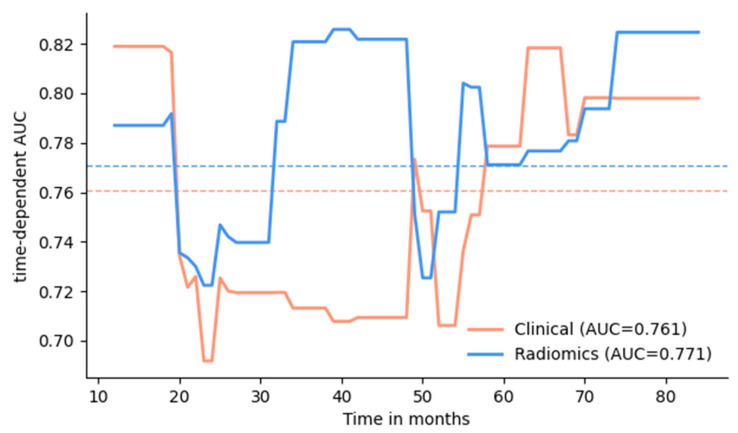
Time-dependent AUC was measured over the period from 1 to 7 years at 1-month intervals, reflecting the performance of predicting overall survival at different time points. The blue line corresponds to the model using radiomics features, and the orange line corresponds to the clinical model. The dotted lines correspond to the mean AUC over the presented time range.

**Figure 4 cancers-14-04449-f004:**
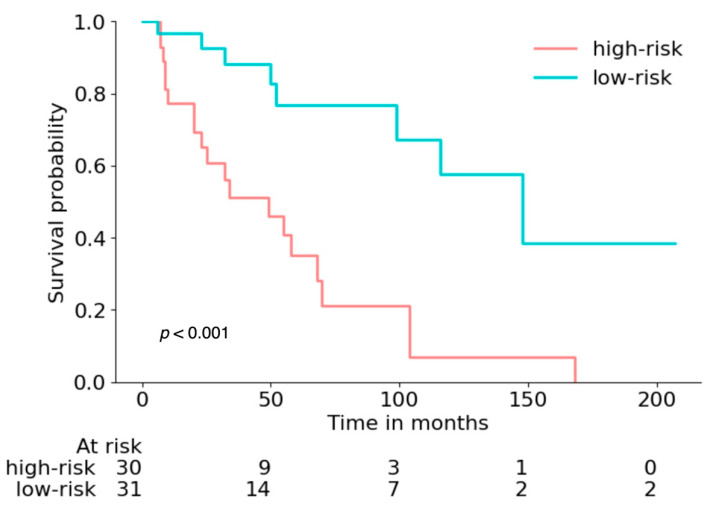
Kaplan–Meyer analysis for the combined model dichotomized into high- and low-risk groups according to the median predicted risk score from the training dataset, evaluated in the test dataset. The log-rank test was used to compare risk groups. The number of subjects at risk is provided below the graph.

**Figure 5 cancers-14-04449-f005:**
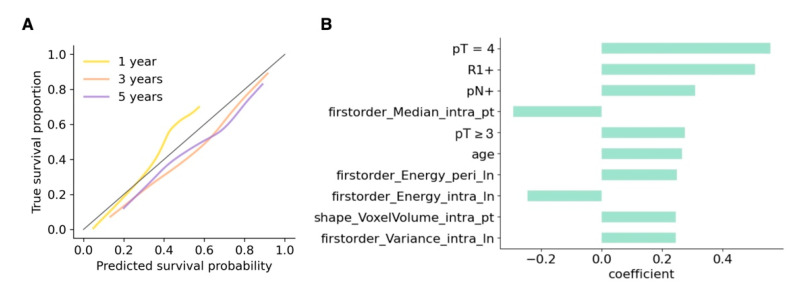
(**A**) Calibration curves for the combined model of predicted survival probability versus true survival proportion for the test dataset at 1, 3 and 5 years. The closer to the diagonal the curves are, the better the calibration. (**B**) The 10 features with the highest coefficient for the combined model correspond to its most important features; R1 = positive surgical margins (positive resection status); intra = imaging features extracted from intratumoral segmentations; peri = imaging features extracted from peritumoral segmentations; pt = primary tumor; ln *=* lymph node.

**Table 1 cancers-14-04449-t001:** Clinical characteristics of the study cohort.

Parameter	Training Set (*n* = 240)	Test Set (*n* = 61)	*p*-Value
Sex:			0.088
Male	168 (70%)	50 (82%)	
Female	72 (30%)	11 (18%)	
Age	69 ± 10	69 ± 11	0.980
T stage:			0.698
pT ≤ 1	35 (15%)	9 (15%)	
pT2	80 (33%)	20 (33%)	
pT3	90 (38%)	22 (36%)	
pT4	34 (14%)	9 (15%)	
pTx	1 (0%)	1 (1%)	
N stage:			0.244
pN0	177 (74%)	50 (82%)	
pN1/pN2	63 (26%)	11 (18%)	
Tumor volume (mL)	10.6 [4.6–24.1]	11.7 [5.5–27.4]	0.649
Resection status (surgical margins):			0.338
R0	207 (86%)	49 (80%)	
R1/R2	33 (14%)	12 (20%)	
Lymphovascular invasion present	71 (30%)	12 (20%)	0.260
Time between CT and surgery (days)	17 [4–35]	17 [3–35]	0.891
Time to recurrence (months)	8 (7–11)	16 (4–33)	0.5
Postopertive complications (Clavien-Dindo)			0.84
low-grade (0–2)	198	52	
high-grade (3–5)	42	9	
Deaths	141	29	0.06
cancer-specific	98 (70%)	14 (48%)	
other cause	38 (36%)	14 (48%)	
unclear/unknown cause	5 (4%)	1 (4%)	

Data reported as mean ± std, or median [IQR], or median (95% CI).

**Table 2 cancers-14-04449-t002:** Results of the analysis (95% confidence intervals in brackets).

Model	AUC (1–7 y)	*p*–Val *	Brier (1–7 y)	*p*–Val *	C–Index
Clinical model	0.761 (0.617–0.874)	ref.	0.185 (0.144–0.232)	ref.	0.722 (0.609–0.823)
Radiomics features:					
intra (BCa + LN)	0.706 (0.552–0.837)	0.90	0.221 (0.179–0.263)	0.50	0.676 (0.549–0.791)
intra + peri (BCa)	0.731 (0.626–0.828)	1.0	0.210 (0.169–0.253)	0.50	0.731 (0.626–0.828)
intra + peri (BCa + LN)	0.771 (0.657–0.869)	0.98	0.202 (0.163–0.243)	0.53	0.737 (0.644–0.822)
Combined model	0.785 (0.648–0.891)	0.69	0.175 (0.129–0.224)	1.0	0.740 (0.632–0.836)

AUC—integrated area under the ROC curve, Brier—integrated Brier score, C-index—concordance index, y—year. * Calculated using the distribution of differences from bootstrapping with 1000 resamples.

**Table 3 cancers-14-04449-t003:** Results of the analysis in terms of AUC and Brier score, measured at 1, 2, 3, 5 and 7 years after surgery.

**Model**	**AUC (1 y)**	**AUC (2 y)**	**AUC (3 y)**	**AUC (5 y)**	**AUC (7 y)**
Clinical model	0.819 (0.647–0.960)	0.692 (0.482–0.885)	0.713 (0.545–0.875)	0.779 (0.622–0.921)	0.798 (0.638–0.924)
Radiomics features:					
intra (BCa + LN)	0.758 (0.604–0.897)	0.712 (0.512–0.925)	0.784 (0.627–0.925)	0.802 (0.642–0.928)	0.820 (0.665–0.945)
intra + peri (BCa)	0.758 (0.545–0.921)	0.672 (0.456–0.859)	0.706 (0.524–0.864)	0.680 (0.493–0.843)	0.711 (0.517–0.876)
intra + peri (BCa + LN)	0.787 (0.650–0.900)	0.722 (0.544–0.869)	0.821 (0.672–0.944)	0.771 (0.601–0.916)	0.825 (0.676–0.950)
Combined model	0.845 (0.688–0.968)	0.769 (0.588–0.918)	0.790 (0.632–0.915)	0.749 (0.571–0.907)	0.803 (0.639–0.907)
**Model**	**Brier (1 y)**	**Brier (2 y)**	**Brier (3 y)**	**Brier (5 y)**	**Brier (7 y)**
Clinical model	0.105 (0.067–0.144)	0.172 (0.120–0.229)	0.202 (0.145–0.266)	0.195 (0.131–0.269)	0.210 (0.136–0.296)
Radiomics features:					
intra (BCa + LN)	0.130 (0.068–0.178)	0.195 (0.153–0.242)	0.214 (0.166–0.267)	0.214 (0.160–0.275)	0.252 (0.167–0.358)
intra + peri (BCa)	0.137 (0.087–0.192)	0.198 (0.151–0.253)	0.228 (0.181–0.286)	0.230 (0.176–0.284)	0.264 (0.184–0.363
intra + peri (BCa + LN)	0.121 (0.082–0.166)	0.186 (0.148–0.250)	0.201 (0.158–0.250)	0.213 (0.161–0.272)	0.243 (0.162–0.344)
Combined model	0.098 (0.051–0.156)	0.160 (0.105–0.226)	0.174 (0.116–0.245)	0.197 (0.130–0.275)	0.205 (0.123–0.294)

AUC—area under the curve, y—year.

## Data Availability

The complete code generated in this study can be shared and made publicly available on request.

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
