# Peer review of "Radiomics for the Prediction of Overall Survival in Patients with Bladder Cancer Prior to Radical Cystectomy"

_cancers, 2022, doi:10.3390/cancers14184449_

Round 1
Reviewer 1 Report
The authors showed that radiomics features in CT before radical cystectomy can predict OS and complement the clinical model. Although not externally validated, this is an important result that demonstrates the potential utility of radiomics features. It is interesting to note that the peritumoral findings were also addressed. However, drawing definitive conclusions about OS from the limited results is a problem.
Despite the primary endpoint being OS, it is difficult for urologists and readers to interpret the results of this study, because the analysis does not include information on patient characteristics of adjuvant or salvage chemotherapy or complications etc. First, the impact on recurrence and cancer-specific survival should be shown.
Although preoperative adjuvant chemotherapy is now the standard of care, it is not included in this study, and it is unclear what specific clinical decision-making improvements will have for this patient population, since the improvements are not dramatic compared to the clinical model.
Since manual segmentation was used in conjunction, it may be an arbitrary voxel volume. Does radiomics model have clinical significance beyond predicting pT stage and pN status? What are the results of second-order features for intra+peri?
Minor
Fig1 typo *Identification
Fig5B Please write down the features without abbreviation.
Reviewer 2 Report
General impression:
The authors report a thoroughly conducted and well-described analysis. All results are comprehensible, and statistical analyses are sound and meaningful. The selection process for eligible patients and datasets is sound and transparent. Descriptions of the methods are concise but comprehensible and sufficient. Descriptive statistics are helpful. The clinical question is clear and valid.
Unfortunately, I could not access and evaluate the Supplemental Material.
Major remark:
I appreciate the extensive descriptive comparison of model performance between the clinical model and the combined model. In my opinion, demonstrating that radiomics features add value to the established clinical models is a key element of any radiomics publication because time-consuming processes such as manual segmentation and feature processing should be justified by superior overall accuracy. At least one key metric should therefore be compared statistically between the models to demonstrate if the combined model is truly superior to the clinical model. If both models show similar predictive accuracy (which I assume), this should be transparent to the reader. Such an analysis would also help to evaluate the authors’ statement in line 240-242. This statistical comparison should be appropriate for the model structure, e.g. if the models are not nested, the authors may consider the Akaike information criterion (AIC) and relative likelihood. I would recommend including the key performance metric(s) for the clinical model and the result of this model comparison in the abstract because this is one of the key points for the reader. Unless radiomics features significantly improve overall model accuracy, there is arguably no clinical value for them.
Minor comments:
Scheme 1: Are the median values or mean values given?
Scheme 2: The number of features with non-zero coefficients in the combined model is far higher than the sum of features in the radiomics model + clinical model. Are the numbers correct? If yes, is there an explanation? Please provide a full list of features in the final models (e.g., in the supplemental material).
Please describe briefly the steps of feature processing that were performed, such as resampling, interpolation, discretization etc. Did you check if your process of feature extraction was compliant with the IBSI standard (i.e., do the extracted features with your pipeline comply with the reference ranges set by the IBSI for their dataset [pubmed-ID 32154773]? Ideally, the authors would provide an IBSI compliance check spreadsheet.
Line 293: “[…] … prognostic value of neoadjuvant chemotherapy … […]”. Do the authors mean “adjuvant”?
Conclusions (line 297): “Radiomics supports the survival stratification […]”. What to the authors mean with “supports”? Strictly speaking, the authors have not demonstrated that radiomics improve predictive accuracy significantly compared to the established clinical parameters.
Round 2
Reviewer 1 Report
The authors have revised the manuscript appropriately and the content is now clearer. The study suggests a little utility, though perhaps not enough, regarding radiomics in bladder cancer. We look forward to the continuation of larger, objectively meaningful studies in the future.
Reviewer 2 Report
All of my comments have been adressed.